# Outcomes of Duct-to-Mucosa vs. Invagination Pancreatojejunostomy: Toward a Personalized Approach for Distal Pancreatic Stump Anastomosis in Central Pancreatectomy?

**DOI:** 10.3390/jpm13050858

**Published:** 2023-05-20

**Authors:** Traian Dumitrascu, Irinel Popescu

**Affiliations:** 1Center of General Surgery and Liver Transplant, Fundeni Clinical Institute, Carol Davila University of Medicine and Pharmacy, 022328 Bucharest, Romania; 2Center of General Surgery and Liver Transplant, Fundeni Clinical Institute, Titu Maiorescu University, 022328 Bucharest, Romania; irinel.popescu2020@gmail.com

**Keywords:** central pancreatectomy, distal pancreatic stump, duct-to-mucosa anastomosis, jejunal invagination technique, morbidity

## Abstract

(1) Background: The jejunum is primarily used for distal pancreatic stump anastomoses after central pancreatectomy (CP). The study aimed to compare duct-to-mucosa (WJ) and distal pancreatic invagination into jejunum anastomoses (PJ) after CP. (2) Methods: All patients with CP and jejunal anastomoses (between 1 January 2002 and 31 December 2022) were retrospectively assessed and compared. (3) Results: 29 CP were analyzed: WJ—12 patients (41.4%) and PJ—17 patients (58.6%). The operative time was significantly higher in the WJ vs. PJ group of patients (195 min vs. 140 min, *p* = 0.012). Statistically higher rates of patients within the high-risk fistula group were observed in the PJ vs. WJ group (52.9% vs. 0%, *p* = 0.003). However, no differences were observed between the groups regarding the overall, severe, and specific postpancreatectomy morbidity rates (*p* values ≥ 0.170). (4) Conclusions: The WJ and PJ anastomoses after CP were comparable in terms of morbidity rates. However, a PJ anastomosis appeared to fit better for patients with high-risk fistula scores. Thus, a personalized, patient-adapted technique for the distal pancreatic stump anastomosis with the jejunum after CP should be considered. At the same time, future research should explore gastric anastomoses’ emerging role.

## 1. Introduction

Central pancreatectomy (CP) is a rarely performed [1,2] and controversial type of pancreatic resection [3] that is sometimes used as an alternative to distal pancreatectomy mainly for certain benign and low-grade malignant tumors of the pancreatic body and isthmus [4]. The principal advantage of a CP over a distal pancreatectomy is a better preservation of postoperative pancreatic functions, particularly the endocrine one [5,6,7,8]. However, a CP is associated with high morbidity rates (higher than distal pancreatectomies) [5,6,7,9] and exceptionally high rates of postoperative pancreatic fistulae (POPF) [2,3]. Thus, even in large series of patients from high-volume centers, the POPF rates after CP are around 45.3–63% [10,11,12]. A potential explanation for the high rates of POPF is the presence of two remaining pancreatic stumps (proximal and distal) [11,13] and the fact that the indications for CP are mainly represented by benign cystic or neuroendocrine pancreatic tumors [7,11], a situation widely considered to be associated with increased rates of POPF [14,15]. Furthermore, most patients with CP have soft pancreas texture and small Wirsung ducts [11], two determinant risk factors for POPF development [14,15]. Nevertheless, the distal pancreatic stump anastomosis after CP remains a significant source of postoperative complications, namely POPF.

The jejunum is primarily used for distal pancreatic stump anastomoses after CP [1,4,8,11,12,16]. However, in a few centers, an anastomosis with the stomach is preferred for the distal pancreatic stump after CP [10,17,18], especially in a minimally invasive approach [19,20].

Various reconstruction techniques have been proposed for the distal pancreatic stump to reduce POPF rates after CP and pancreaticoduodenectomy (PD) for both anastomoses with the jejunum and the stomach [21,22,23,24,25,26,27]. Although many studies are comparing the outcomes of different types of anastomoses of the distal pancreatic stump after PD [21,22,23,24,25,26,27,28,29,30,31,32], for CP, the data are scarce. Thus, to date, only three studies have compared the distal pancreatic stump anastomoses with the jejunum vs. the stomach after CP [12,33,34], and one study has compared pancreatojejunostomy with end-to-end Wirsung duct anastomoses [35]. Interestingly, no study has compared different types of distal pancreatic stump anastomoses with the jejunum after CP.

The present study aimed to compare duct-to-mucosa Wirsungo-jejunal anastomoses (WJ) and distal pancreatic invagination into the jejunum anastomoses (PJ) in a single-center series of patients with CP.

## 2. Materials and Methods

### 2.1. Patients and Surgical Technique

The data of all patients with CP performed between 1 January 2002 and 31 December 2022 in our surgical center were retrospectively assessed from a prospectively maintained electronic database.

Our technique for CP was previously described elsewhere [36,37]. Patients with a distal pancreatic stump anastomosis with the stomach were excluded. A comparison regarding outcomes between the distal pancreatic stump anastomoses with the stomach and distal pancreatic stump anastomoses with the jejunum would have added value to the present study but it was not feasible because the number of patients in the group of patients with CP and distal pancreatic stump anastomoses with the stomach was very small.

The patients were split into two groups: patients with duct-to-mucosa Wirsungo-jejunal anastomoses (WJ group) and patients with distal pancreatic stump invagination into the jejunum anastomoses (PJ group). The invagination technique was constructed into an end-to-side single layer of interrupted nonabsorbable sutures between the pancreatic parenchyma and capsule and the full-thickness jejunum. The duct-to-mucosa technique consisted of a double layer of interrupted nonabsorbable sutures: the inner layer, an end-to-side anastomosis of the Wirsung duct with the full-thickness jejunum; the outer layer, sutures between the pancreatic parenchyma and capsule and the seromuscular jejunum. The choice for the distal pancreatic stump anastomoses was not standardized, and it was made mainly according to the surgeon’s expertise and preference. The surgeons who performed the surgical procedures were familiar with both anastomotic techniques and usually, when the Wirsung duct diameter was large enough, a duct-to-mucosa technique was first considered as the reconstruction method for the distal pancreatic stump after CP.

The patients’ pre, intra, and postoperative data were comparatively assessed. The preoperative data included sex, age, body mass index, smoking and alcohol abuse, cardiovascular comorbidities, associated diabetes mellitus and chronic pancreatitis, and the American Society of Anesthesiologists and the Karnofsky scores. The intraoperative parameters included estimated blood loss, operative time, tumor diameter, pancreas texture, the diameter of the Wirsung duct, and associated procedures and pathology.

### 2.2. Definition of Outcomes

The early morbidity was defined as in-hospital complications. The Dindo classification and grading of postoperative complications were used [38]. A complication grade 3 or 4 was considered a severe complication. For the specific postpancreatectomy complications such as POPF [39], postoperative hemorrhage (PPH) [40], and delayed gastric emptying (DGE) [41], the definitions and grading of the International Study Group for Pancreatic Surgery were used. The postoperative mortality was assessed at 90 days.

For the long-term outcomes, the patients were periodically followed till death occurrence or the last follow-up update (1 February 2023).

### 2.3. Statistics

Data are expressed as number (percentage) for the categorical variables and as median (range) for the continuous variables, except for the overall survival time where the medians were not reached; for the overall survival time, data are presented as mean ± standard deviation. The comparisons between the groups were made using the Mann–Whitney test (for continuous variables) and Fisher’s exact two-tailed test (for categorical variables). Kaplan–Meier curves were used for the overall survival time, and the groups’ comparisons were made using the long-rank test. *p* values less than 0.05 were considered statistically significant.

## 3. Results

During the analyzed period, in our surgical center, 34 CPs were performed. The analyses excluded five patients (14.7%) with distal pancreas stump anastomoses with the stomach. Thus, the cohort included 29 patients with CP split into two groups: the WJ group—12 patients (41.4%) and the PJ group—17 patients (58.6%).

### 3.1. Demographics and Preoperative Parameters

No statistically significant differences were observed between the WJ and PJ groups of patients regarding the primary demographics and preoperative parameters (*p* values ≥ 0.105), as shown in Table 1.

### 3.2. Intraoperative Data

All CPs included in the present analyses were performed through an open approach. The preservation of the splenic vessels and spleen was performed in all analyzed patients.

No statistically significant differences were observed between the WJ and PJ groups of patients regarding the indication for CP from the point of view of pathology (including neuroendocrine pathology), associated procedures, tumor diameter, and estimated blood loss (*p* values ≥ 0.138), as shown in Table 2. No patient in the present cohort required intraoperative blood transfusions.

The operative time was statistically significant higher in the WJ group of patients, compared with the PJ group of patients (195 min, range: 120–480 min vs. 140 min, range: 100–205 min, *p* = 0.012) (Table 2). Furthermore, although there were no statistically significant differences between the groups regarding the percent of patients with a soft pancreas texture (83.3% vs. 94.1%, *p* = 0.553, ns), the Wirsung duct diameter was statistically significantly higher in the WJ group of patients, compared with the PJ group of patients (3 mm, range: 2–3 mm vs. 1 mm, range: 1–6 mm, *p* < 0.0003) (Table 2).

Of note, statistically significant differences were observed between the groups regarding the pancreatic fistula risk scores [42] (WJ group: 5 points, range: 3–6 points vs. PJ group: 7 points, range: 2–7 points, *p* < 0.001) (Figure 1), with statistically significant higher rates of patients within the high-risk fistula group [42] in the PJ group of patients, compared with the WJ group of patients (52.9% vs. 0%, *p* = 0.003) (Table 2).

### 3.3. Early Postoperative Outcomes

No statistically significant differences were observed between the WJ and PJ groups of patients regarding the overall morbidity (including comprehensive complication index [43]), severe morbidity, clinically relevant POPF, DGE and PPH, postoperative blood transfusions and hospital stays, postoperative abdominal drainage time, relaparotomy for complications, and 90-day mortality rates (*p* values ≥ 0.170) (Table 3).

### 
3.4. Late Postoperative Outcomes


No statistically significant differences were observed between the WJ and PJ groups of patients regarding the mean overall survival time (232 ± 18 months vs. 215 ± 10 months, *p* = 0.780, ns), as shown in Figure 2. However, it is worth mentioning that there was a statistically significant difference between the WJ and PJ groups of patients for the median follow-up time (203 months, range: 28–251 months vs. 157 months, range: 46–225 months, *p* = 0.019) (Table 4). The 20-year survival rates were 94% for the PJ group of patients and 92% for the WJ group. In the PJ group of patients, a patient resected for metastatic melanoma to the pancreas died with peritoneal recurrence 48 months after CP. In the WJ group of patients, two deaths were observed: one patient resected for metastatic colon cancer to the pancreas died with peritoneal recurrence 28 months after CP; in contrast, the other patient, resected for a neuroendocrine tumor died 176 months after CP, but the cause of death was not related to the pancreas pathology.

Regarding the functional postoperative results, no statistically significant differences were observed between the WJ and PJ groups of patients regarding the new-onset or worsening diabetes mellitus rates. None of the patients in the present cohort developed clinically relevant exocrine pancreatic insufficiencies (Table 4).

## 4. Discussion

Data about the outcomes of different techniques for distal pancreatic stump anastomoses are coming mainly from patients with PD [21,22,23,24,25,26,27,28,29,30,31,32,44,45,46], while a minority of studies comparatively explore patients with CP [12,33,34,35].

Mainly, there are two types of jejunal anastomoses of the distal pancreatic stump after PD or CP: duct-to-mucosa and invagination techniques [11,25,28,31,42,47] (Figure 3 and Figure 4). The invagination technique is easier to perform. However, it might be associated with jejunal or pancreatic stump ischemia in the early outcomes and pancreatic stump necrosis, infection, and stenosis in the long-term outcomes due to a direct exposure to the digestive juice. A duct-to-mucosa technique could promote tissue healing but requires more outstanding surgical expertise. Furthermore, a duct-to-mucosa anastomosis might lower the risk of infection and hemorrhage of the distal pancreatic stump by preventing the pancreatic stump from being eroded by digestive juice. However, a duct-to-mucosa anastomosis might be extremely challenging in the setting of a small Wirsung duct and a soft pancreas texture [24].

The current evidence does not favor any specific technique for the distal pancreatic stump anastomoses with the jejunum after PD [21,22,23,24,25,26,28,29,30,31,44,47,48]. However, a few studies have suggested the association of the duct-to-mucosa technique with lower POPF rates, compared with the invagination technique after PD [46], particularly for patients with dilated Wirsung duct [25]. The invagination technique appears to be the first choice for a soft pancreas texture based on the results of a few studies [21,27,45,49]. In contrast, other studies associated the invagination technique with increased rates of POPF and mortality in patients with PD and soft pancreas texture [50].

In the present study, there were no statistically significant differences between the WJ and the PJ group of patients regarding the overall and severe morbidity, clinically relevant POPF, DGE and PPH, postoperative blood transfusions and hospital stays, postoperative abdominal drainage time, relaparotomy for complications, and 90-day mortality (*p* values ≥ 0.170), as shown in Table 3. Similar results were previously reported for patients with PD [21,22,23,24,25,26,28,29,30,31,44,47,48]. One study associated PJ anastomoses with reduced hospital stays and postoperative abdominal drainage times in patients with PD, compared with WJ anastomoses [45]. Other studies associated WJ anastomoses with reduced severe complications, PPH rates, and hospital stays in patients with PD, compared with PJ anastomoses [32,46]. It is worth mentioning that in the present study, there were no differences between the WJ and PJ groups of patients regarding a few preoperative factors that could potentially influence the postoperative complication rates after CP (Table 1). Thus, few studies have shown that age and body mass index are independent predictors of morbidity after CP [11,51,52].

Considering the lack of statistically significant differences in postoperative complications between the WJ and the PJ group of patients (Table 3), one might conclude that a pancreatic surgeon can equally choose any of the two techniques to treat distal pancreatic stumps after CP. However, considering the statistically significant higher rates of patients within the high-risk fistula group in the PJ group of patients, compared with the WJ group of patients (52.9% vs. 0%, *p* = 0.003) (Table 2), it might suggest a personalized, patient-adapted approach for the distal pancreatic stump anastomoses after CP. Thus, the results of the present study appear to favor a PJ over WJ anastomosis for a distal pancreatic stump in patients with CP and high-risk pancreatic fistula scores. In contrast, a WJ anastomosis appears to be a better choice over PJ in patients with a dilated Wirsung duct. Similar data were reported in a recent study by Kone and coworkers but for patients with PD [25].

In the present study, the operative time was statistically significantly higher in the WJ group of patients, compared with the PJ group after CP (195 min vs. 140 min, *p* = 0.012) (Table 2). A recent meta-analysis showed similar results but for patients with PD [26]. However, another study, including many patients, associated duct-to-mucosa anastomoses with lower operative times, compared with the invagination technique after PD [25]. Nevertheless, a meta-analysis by Lyu and coworkers found no significant differences in the operative times between the two techniques in patients with PD [22].

The present study found no differences between the WJ and PJ groups of patients regarding the endocrine and exocrine postoperative functions after CP (Table 4). Similar data were previously reported for patients with PD [29,48].

The present study has a few limitations: its retrospective design, the heterogeneity of the surgeon’s experience for CP, the choice for the surgical technique of the distal pancreatic stump anastomosis being mainly made based on the surgeon’s preference and expertise and not on objective criteria, and the small number of analyzed patients over a relatively long period. Furthermore, the potential benefit of pancreatogastric anastomoses could not be explored due to the small number of patients with distal pancreatic stump anastomoses with the stomach after CP in our center.

Several meta-analyses comparing the outcomes of pancreatojejunostomy with pancreatogastrostomy after PD have reached conflicting results. A few studies did not find any significant differences between the two surgical techniques regarding the incidence of postoperative complications, including POPF and PPH [22,23]. However, other studies suggested that pancreatogastrostomy might be superior to pancreatojejunostomy in preventing clinically relevant POPF [26,53,54]. A recent multicentric study including many patients with PD associated pancreatogastrostomy with higher rates of POPF, compared with the pancreatojejunostomies [32]. Nevertheless, a recent study suggested a tailored approach for distal pancreatic stump anastomosis after PD: a duct-to-mucosa jejunal anastomosis is the first option in patients with a hard pancreatic texture and dilated Wirsung duct. In contrast, for patients with a soft pancreas texture and small Wirsung duct, an invagination pancreatogastrostomy should be preferred [55]. Thus, there is a potential emerging role for using the stomach, particularly for patients with PD and high-risk fistula scores [55].

For CP, there are a limited number of studies comparing gastric and jejunal anastomoses, and the results appear to be at odds with those obtained in patients with PD. For example, in a recent study, the use of the stomach for the distal pancreatic stump anastomosis was associated with statistically significant higher rates of POPF and abdominal collections after CP, compared with the anastomoses with the jejunum (80% vs. 48.4%, *p* = 0.004 and 48% vs. 27.5%, *p* = 0.046, respectively) [12]. Statistically significant higher rates of POPF were also reported in two other studies in patients with CP and distal pancreatic stump anastomoses with the stomach, compared with jejunal anastomoses (71.4% vs. 18.2%, *p* = 0.014 and 76.6% vs. 37.5%, *p* = 0.003, respectively) [33,34]. However, one recent study including many patients with CP associated invaginating pancreatogastrostomy with lower rates of POPF after CP, compared with the duct-to-mucosa techniques [11].

## 5. Conclusions

CP is a complex surgical procedure associated with high morbidity rates and exceptionally high rates of POPF. The distal pancreatic stump anastomoses represent an essential source of morbidity after CP. Although there were no significant differences in morbidity rates between WJ and PJ anastomoses after CP, for patients with high-risk fistula scores, a PJ anastomosis appeared to fit better. Thus, a personalized, patient-adapted technique for the distal pancreatic stump anastomosis with the jejunum after CP should be considered. At the same time, future research should explore gastric anastomoses’ emerging role.

## Figures and Tables

**Figure 1 jpm-13-00858-f001:**
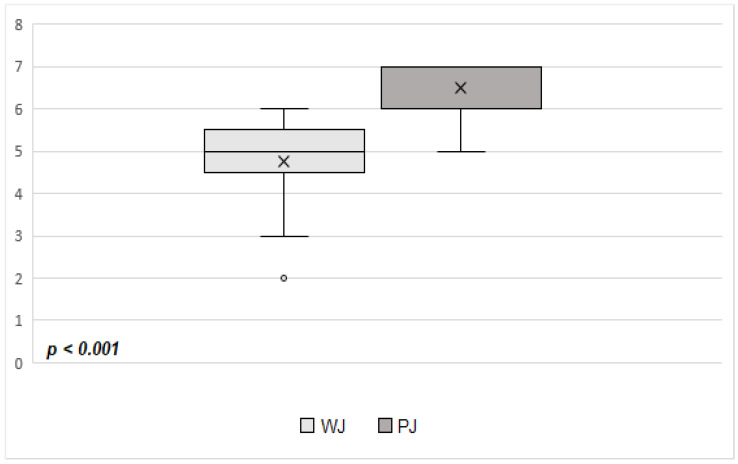
Comparative analysis of fistula risk scores [42] between the WJ and the PJ groups of patients with CP (12 patients vs. 17 patients) (WJ—duct-to-mucosa distal pancreatic stump anastomoses with the jejunum; PJ—invagination distal pancreatic stump anastomoses with the jejunum; CP—central pancreatectomy).

**Figure 2 jpm-13-00858-f002:**
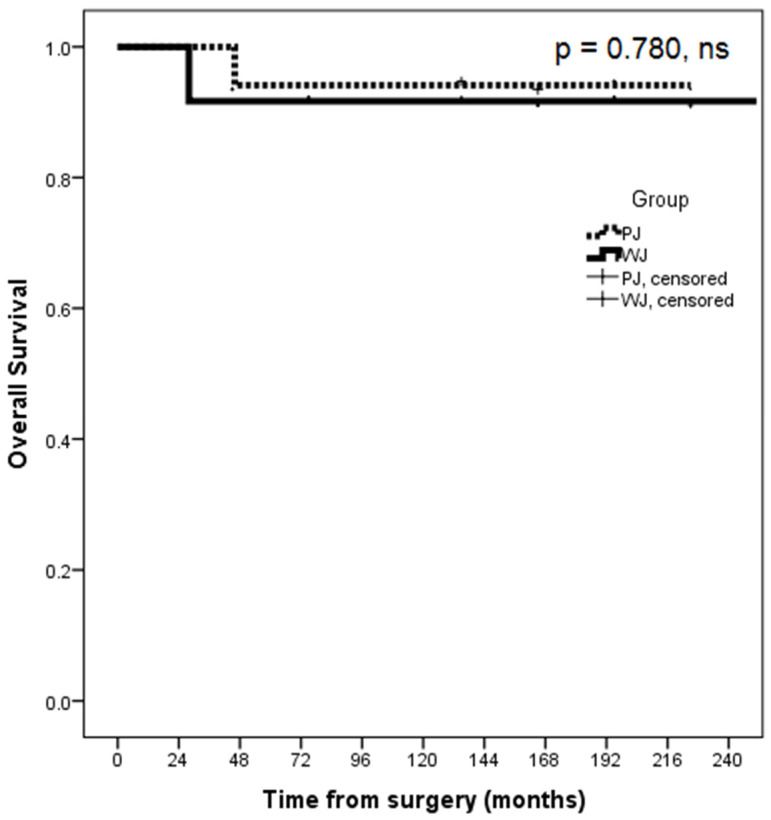
Comparative Kaplan–Meier curves of the mean overall survival time between the WJ and the PJ groups of patients with CP (12 patients vs. 17 patients) (WJ—duct-to-mucosa distal pancreatic stump anastomoses with the jejunum; PJ—invagination distal pancreatic stump anastomoses with the jejunum; CP—central pancreatectomy).

**Figure 3 jpm-13-00858-f003:**
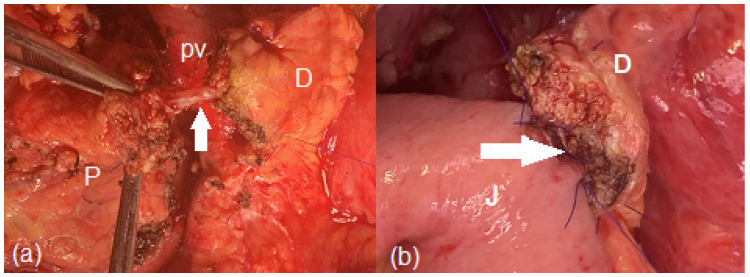
Intraoperative aspects of the duct-to-mucosa anastomosis of the distal pancreatic stump after central pancreatectomy: (**a**) before the anastomosis—the white arrow marks the Wirsung duct; (**b**) during the anastomosis—the white arrow marks the duct-to-mucosa anastomosis with interrupted sutures, the posterior layer completed (P—pancreatic head; D—distal pancreatic stump; J—jejunum; pv—portal vein).

**Figure 4 jpm-13-00858-f004:**
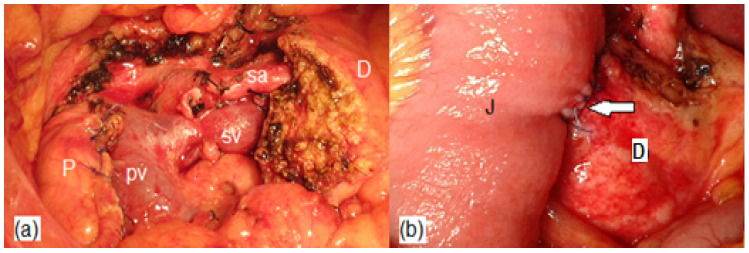
Intraoperative aspects of the jejunal invagination anastomosis of the distal pancreatic stump after central pancreatectomy: (**a**) before the anastomosis, no enlarged Wirsung duct; (**b**) after the anastomosis was completed—the white arrow marks the jejunal invagination of the distal pancreatic stump, end-to-side (P—pancreatic head; D—distal pancreatic stump; J—jejunum; pv—portal vein; sv—splenic vein; sa—splenic artery).

**Table 1 jpm-13-00858-t001:** Comparative analyses of demographics and preoperative parameters between the WJ and the PJ groups of patients with CP (12 patients vs. 17 patients).

Parameter	WJ Group (12 Patients)	PJ Group (17 Patients)	*p* Value
Age, years	51 (14–71)	36 (16–66)	0.412, ns
Female sex	7 patients (58.3%)	12 patients (70.6%)	0.694, ns
Body mass index, kg/m^2^	29.5 (18.5–42)	22 (18.5–36.5)	0.126, ns
Smoking	5 patients (41.7%)	4 patients (23.5%)	0.422, ns
Alcohol abuse	2 patients (16.7%)	1 patient (5.9%)	0.553, ns
Cardiovascular comorbidities	6 patients (50%)	3 patients (17.6%)	0.105, ns
Diabetes mellitus	2 patients (16.7%)	2 patients (11.8%)	1, ns
Chronic pancreatitis	1 patient (8.3%)	1 patient (5.9%)	1, ns
ASA score ≥3	4 patients (33.3%)	3 patients (17.6%)	0.402, ns
Karnofsky score, points	90 (80–100)	90 (80–100)	1, ns

WJ—duct-to-mucosa distal pancreatic stump anastomoses with the jejunum; PJ—invagination distal pancreatic stump anastomoses with the jejunum; ASA—American Society of Anesthesiologists; CP—central pancreatectomy.

**Table 2 jpm-13-00858-t002:** Comparative analyses of intraoperative parameters between the WJ and the PJ groups of patients with CP (12 patients vs. 17 patients).

Parameter	WJ Group (12 Patients)	PJ Group (17 Patients)	*p* Value
Associated procedures	2 patients (16.7%)	2 patients (11.8%)	1, ns
Soft pancreas texture	10 patients (83.3%)	16 patients (94.1%)	0.553, ns
Wirsung duct diameter, mm	3 (2–3)	1 (1–6)	<0.0003
Operative time, min	195 (120–480)	140 (100–205)	0.012
Estimated blood loss, ml	125 (50–500)	50 (50–300)	0.138, ns
Malignant pathology	2 patients (16.7%)	2 patients (11.8%)	1, ns
Neuroendocrine pathology	2 patients (16.7%)	5 patients (29.4%)	0.205, ns
Tumor diameter, cm	3 (2–14)	2.7 (1.1–6)	0.347, ns
Pancreatic Fistula Risk Score [42], points	5 (3–6)	7 (2–7)	<0.001
High-pancreatic fistula risk group [42]	0 patients (0%)	9 patients (52.9%)	0.003

WJ—duct-to-mucosa distal pancreatic stump anastomoses with the jejunum; PJ—invagination distal pancreatic stump anastomoses with the jejunum; CP—central pancreatectomy.

**Table 3 jpm-13-00858-t003:** Comparative analyses of early postoperative outcomes between the WJ and the PJ groups of patients with CP (12 patients vs. 17 patients).

Parameter	WJ Group (12 Patients)	PJ Group (17 Patients)	*p* Value
Overall complications	8 patients (66.7%)	10 patients (58.8%)	0.716, ns
Severe complications (i.e., grade 3–4 Dindo)	4 patients (33.3%)	3 patients (17.6%)	0.402, ns
90-day mortality	0 patients (0%)	0 patients (0%)	1, ns
Clinically relevant POPF (i.e., grade B–C)	5 patients (41.7%)	7 patients (41.2%)	1, ns
Clinically relevant DGE (i.e., grade B–C)	2 patients (16.7%)	3 patients (17.6%)	1, ns
Clinically relevant PPH (i.e., grade B–C)	2 patients (16.7%)	2 patients (11.8%)	1, ns
Postoperative blood transfusions for complications	2 patients (16.7%)	3 patients (17.6%)	1, ns
Relaparotomy for complications	4 patients (33.3%)	3 patients (17.6%)	0.402, ns
Postoperative hospital stays, days	20 (6–45)	12 (7–32)	0.170, ns
Postoperative abdominal drainage time, days	10 (5–44)	5 (5–28)	0.373, ns
Comprehensive Complication Index [43]	8.7 (0–56.1)	8.7 (0–47.7)	0.535, ns

WJ—duct-to-mucosa distal pancreatic stump anastomoses with the jejunum; PJ—invagination distal pancreatic stump anastomoses with the jejunum; POPF—postoperative pancreatic fistula; DGE—delayed gastric emptying; PPH—post pancreatectomy hemorrhage; CP—central pancreatectomy.

**Table 4 jpm-13-00858-t004:** Comparative analyses of late postoperative outcomes between the WJ and the PJ groups of patients with CP (12 patients vs. 17 patients).

Parameter	WJ Group (12 Patients)	PJ Group (17 Patients)	*p* Value
Follow-up time, months	203 (28–251)	157 (46–225)	0.019
Overall survival time *, months	232 ± 18	215 ± 10	0.780, ns
New-onset or worsening diabetes mellitus	2 patients (16.7%)	2 patients (11.8%)	1, ns
Clinically relevant postoperative exocrine pancreatic insufficiency	0 patients (0%)	0 patients (0%)	1, ns

WJ—duct-to-mucosa distal pancreatic stump anastomoses with the jejunum; PJ—invagination distal pancreatic stump anastomoses with the jejunum; * Data are expressed as mean ± standard deviation because the medians were not reached.

## Data Availability

The datasets generated during and/or analyzed during the current study are available from the corresponding author on reasonable request.

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
