# Peer review of "Outcomes of Duct-to-Mucosa vs. Invagination Pancreatojejunostomy: Toward a Personalized Approach for Distal Pancreatic Stump Anastomosis in Central Pancreatectomy?"

_jpm, 2023, doi:10.3390/jpm13050858_

Round 1
Reviewer 1 Report
I read with interest the current paper by Dumitrascu and Popescu on two different anastomoses' techniques for central pancreatectomy.
It is clearly written in decent English.
I have two suggestions for improvement:
1. Please elaborate on the decision process of choosing the WJ or PJ during the operation. Do the surgeons in this study perform both? Or are there specific surgeons who only perform WJs or PJs. This might influence the results. If a surgeon performs both WJs and PJs, when and why does he or she decides to use which anastomoses.
2. It would enhance the paper if you could provide illustrations that visualise the described anastomoses.
Author Response
Response to Reviewer 1
- Please elaborate on the decision process of choosing the WJ or PJ during the operation. Do the surgeons in this study perform both? Or are there specific surgeons who only perform WJs or PJs. This might influence the results. If a surgeon performs both WJs and PJs, when and why does he or she decides to use which anastomoses.
Response 1: The surgeons who performed the surgical procedures were familiar with both anastomotic techniques and, usually, when the Wirsung duct diameter was large enough, a duct-to-mucosa technique was first considered as the reconstruction method for the distal pancreatic stump after CP.
- It would enhance the paper if you could provide illustrations that visualize the described anastomoses.
Response 2: We have introduced two new figures showing the two types of anastomoses.
Reviewer 2 Report
The Authors performed valuable study about WJ vs PJ after CP in the long period albeit with limited cases. The minor concerns are as follows:
1. In Line 105-106, 119, 147, why do the Authors described range of p values?
2. Wirsung duct diameter was significant higher in the WJ group. Please explain the reasons and the Authors’ experience in proper paragraphs.
3. There are redundant spaces in the manuscript like Line 196. Please check the whole manuscript and delete them.
There are redundant spaces in the manuscript like Line 196. Please check the whole manuscript and delete them.
Author Response
Response to Reviewer 2
- In Line 105-106, 119, 147, why do the Authors describe range of p values?
Response 1: We have provided the p values just to show the non-statistical significance, but the p values for each parameter is shown in Tables.
- Wirsung duct diameter was significant higher in the WJ group. Please explain the reasons and the Authors’ experience in proper paragraphs.
Response 2: Whenever a Wirsung duct was considered large enough, a duct-to-mucosa was the first option. That is the explanation for this finding.
- There are redundant spaces in the manuscript like Line 196. Please check the whole manuscript and delete them.
Response 3: We have checked the manuscript and eliminate the redundant spaces.